## Reply

 

**Cite this article:** van Casteren A et al. 2019 Metallic proxies remain unsuitable for assessing the mechanics of microwear formation: reply to comment on van Casteren et al. (2018). R. Soc. open sci. **6**: 190572.

**Subject Category:**
Biology (whole organism)

biomaterials/evolution/palaeontology

**Author for correspondence:**
Adam van Casteren
e-mail: adam.vancasteren@gmail.com

The accompanying comment can be viewed at
http://dx.doi.org/10.1098/rsos.181376.

# Metallic proxies remain unsuitable for assessing the mechanics of microwear formation: reply to comment on van Casteren et al. (2018)

Adam van Casteren[1], Peter W. Lucas[2], David S. Strait[1], Shaji Michael[3], Nick Bierwisch[4], Norbert Schwarzer[4], Khaled J. Al-Fadhalah[5], Abdulwahab S. Almusallam[6], Lidia A. Thai[7], Sreeja Saji[3], Ali Shekeban[7] and Michael V. Swain[8]

[1]Department of Anthropology, Washington University in St Louis, Campus Box 1114, One Brookings Drive, St Louis, MO 63130, USA
[2]Smithsonian Tropical Research Institute, Luis Clement Avenue, Building 401 Tupper Balboa Ancon, Panama, Republic of Panama
[3]Department of Bioclinical Sciences, Faculty of Dentistry, Kuwait University, PO Box 24923, Safat 11310, Kuwait
[4]Saxonian Institute of Surface Mechanics SIO, Tankow 2, 18569 Ummanz/Rügen, Germany
[5]Department of Mechanical Engineering, [6]Department of Chemical Engineering, and
[7]Nanotechnology Research Facility, College of Engineering and Petroleum, Kuwait University, PO Box 5969, Safat 13060, Kuwait
[8]Department of Bioengineering, Don State Technical University, Rostov-on-Don, Russia

AvC, 0000-0002-2993-8874; PWL, 0000-0001-5286-9101

## 1. Introduction

A debate has recently developed concerning mechanisms of dental microwear formation. Lucas et al. [1] predicted and demonstrated that phytoliths (microscopic plant silicates) are too soft to wear tooth enamel on initial contact. Rather they should plastically deform (i.e. rub) enamel such that any wear would be the result of fatigue caused by multiple loadings. Xia et al. [2] repeatedly slid macroscopic aluminium and brass balls across enamel and observed seemingly abrasive striations. Because the balls are rounded and both metals measured as softer than enamel, Xia et al. [2] claimed to have falsified the model of Lucas et al. [1]. van Casteren et al. [3] countered that the surfaces of the aluminium balls were coated by a hard, brittle, irregular oxide that would fragment easily and abrade enamel. We also

**Figure 1.** (a) Area of contact plotted against penetration depth, as derived from indenter tip calibration using fused silica for indenter tips used by Xia *et al.* [4] and van Casteren *et al.* [3]. (b) Estimated effective indenter tip radii for the indenters used by Xia *et al.* [4] and van Casteren *et al.* [3]. The equivalent radius $R$ is determined from the expression $A = \pi h_c R$, where $A$ is the contact area and $h_c$, the contact depth of penetration. The plot inserts define both $h_c$ and $h_{\text{trans}}$, which is the depth at which sphere-like contact transitions to a pyramidal (conical) indenter.

found that the brass balls were likely to have been work hardened and may have rubbed rather than directly abraded enamel, and that the high number of sliding trials conducted by Xia *et al.* [2] might have produced abrasion by fatigue. Subsequently, Xia *et al.* [4] criticized the methods and hardness measurements of van Casteren *et al.* [3] and presented results from single slide experiments claiming abrasion. We respond below.

First, it is incontrovertible that the aluminium balls in question are coated by an oxide layer, as shown by energy-dispersive X-ray spectroscopy [2,3], and it is known that aluminium oxide is much harder than aluminium [5]. Furthermore, it is unsurprising that machining processes for producing ball bearings could result in work hardening. These facts alone support our contention that 'metallic proxies are unsuitable for assessing the mechanics of microwear formation' [3, p. 1]. If one wanted to test our assertion that phytoliths cannot directly abrade enamel [1], then the obvious course of action is to slide phytoliths against enamel. In the absence of such experiments, the debate is reduced to materials science and contact mechanics minutiae rather than biologically significant mechanics.

Xia *et al.* [4] cite Hernot *et al.* [6] to argue we used the wrong formula in our hardness calculations. However, the magnitude of error is negligible as our plots are of the contact pressure during initial elastic loading (at only a few nm of depth) and the onset of the first pop-in event, not when sinking-in or pile-up is occurring about the indenter (the condition addressed by Hernot *et al.* [6]). Indeed, Xia *et al.* [4, p. 3] note that, '…the formula [used by van Casteren *et al.* [3]] is accurate only when the indentation process is fully elastic.' Our experimental conditions satisfy this criterion. Moreover, the appropriate measure of a Berkovitch indenter tip when estimating the depth of 'spherical' contact is the effective conical angle, which is 70.3°. Thus, their comment about the depth at which we can accurately infer pressure is invalid.

Ironically, Xia *et al.* [4] measure hardness by relying only on the unloading curve as their basis for interpretation of the contact stresses, yet loading curves for aluminium [2,3] show instances of pop-in behaviour and changes of slope indicative of a hard surface layer. Thus, Xia *et al.* [4] have penetrated the layer they purport to be measuring and have recorded the hardness of the softer, underlying metal.

Xia *et al.* [4] are concerned that the tip radius of our indenter [3] was too large, and they advise calibrating the size of an indenter tip against an area function. Their indenter tip was ostensibly sharper than that used by van Casteren *et al.* [3], but in fact, their own loading curve data and (now corrected) area function indicate their tip is extraordinarily blunt (figure 1). The area curve from van Casteren *et al.* [3] shows that the tip used by us is somewhat blunter than manufacturer's specifications, and we have updated our pressure/hardness data accordingly. Even using the updated calculation (figure 2), pressure on the oxide layer is in the range of enamel after a depth of only approximately 4 nm in the trial whose loading curve shows no sign of pop-in cracks, and the hardness of enamel is certainly exceeded by approximately 9 nm. This is reasonably only a minimum estimate of the oxide hardness, because the pressure is a function of the thin layer flexing on the surface of the softer, underlying metal. Pop-in cracks indicate that the oxide is fracturing soon after, so pressures at greater depth are irrelevant to the hardness of the oxide. van Casteren *et al.* [3] discussed the difficulty inherent in

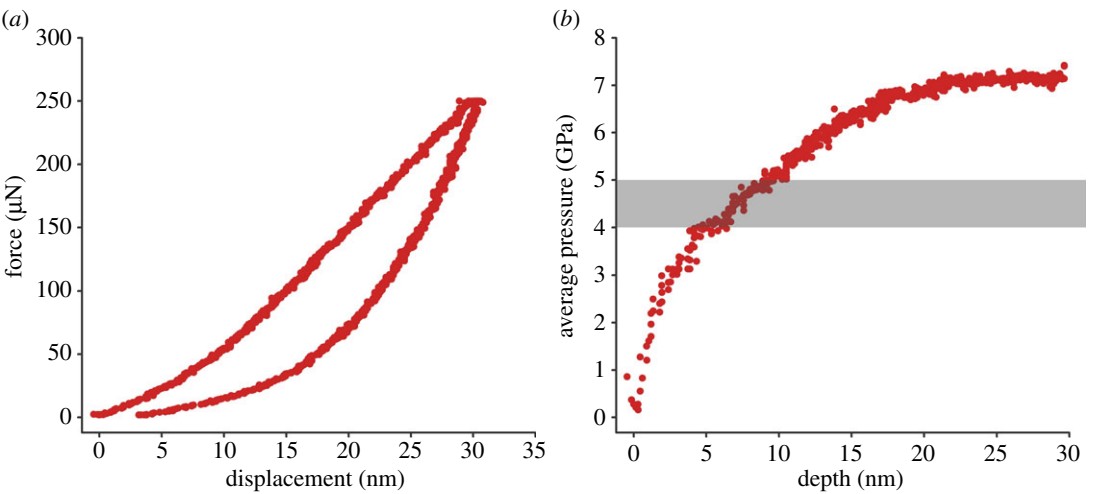

**Figure 2.** (*a*) Example of a force–displacement curve of nanoindentation on the outer surface of an aluminium ball, generated by van Casteren *et al.* [3], showing a steep initial loading curve. This curve showed the least evidence of pop-in events in that study. (*b*) Recalculated average contact pressures for the load–displacement curve in figure 2*a* during loading. These pressures are given by the force at a specific contact depth divided by the contact area at that same depth. During elastic contact with a sphere, the contact depth is half the total depth of penetration. Note that average pressures exceed the hardness of enamel (grey shading) at indentation depths less than $h_{trans}$ (approx. 16 nm) where the tip changes from spherical to pyramidal form for the indenter used by van Casteren *et al.* [3].

measuring the hardness of such thin surfaces, which in and of itself is indicative of the inappropriateness of using metal balls as a proxy for phytoliths. Xia *et al.* [4] took no steps to address this issue.

Xia *et al.* [4] pointed to enamel chips as evidence of abrasive wear caused by their brass single sliding experiments. They do not report the size of these chips, but they can hardly be visualized in their figure 5*e*. Based on their now corrected scale, we surmise that these chips are perhaps several tens of nm in diameter. By contrast, Lucas *et al.* [1] found that sliding quartz dust across enamel produced multiple enamel chips of approximately 2 μm in diameter, but that no such chips were observed when sliding phytoliths across equivalent surfaces. The volume of enamel contained in a 2 μm diameter chip is orders of magnitude greater than that contained in a chip whose diameter is approximately 50–100 nm. Thus, although Xia *et al.* [4] have shown that a brass ball can produce nano-scale damage, they have not demonstrated micro-scale wear. It is the micro-scale that is relevant both to the mechanical model of Lucas *et al.* [1] and to dental microwear analysis generally. van Casteren *et al.* [3] discussed the issue of scale but Xia *et al.* [5] have seemingly ignored the point.

In summary, sliding experiments using metallic balls present little useful data for testing the hypothesis of Lucas *et al.* [1]. An understanding of how biological materials affect teeth should be based on analyses of biological materials.

Data accessibility. The datasets supporting this article have been uploaded as part of the electronic supplementary material.

Authors' contributions. All authors contributed equally to the analysis of data and writing of this paper. All authors read the draft and approved submission.

Competing interests. We declare we have no competing interests.

Funding. We acknowledge support from Kuwait University General Facilities Projects SRUL 1/14, GE 01/07 and also Kuwait University project grants DB 01/12.

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
