## [Reviewer comments · Royal Society Open Science]

Review History

RSOS-190572.R0 (Original submission)

Review form: Reviewer 1 (Mikael Fortelius)

Is the manuscript scientifically sound in its present form?

Yes

Are the interpretations and conclusions justified by the results?

Yes

Is the language acceptable?

Yes

Is it clear how to access all supporting data?

Not Applicable

Do you have any ethical concerns with this paper?

No

Have you any concerns about statistical analyses in this paper?

I do not feel qualified to assess the statistics

Recommendation?

Accept with minor revision (please list in comments)

Comments to the Author(s)

This is a hard one to review, mostly because the discussion has moved into a highly specialised realm of metallic engineering where few will be able to assess the arguments with confidence. It might be tempting to agree with the punch line that “sliding experiments using metallic balls remain irrelevant to testing the hypothesis of Lucas et al.” – but while there are clearly technical complications to be discussed it is by no means obvious that such experiments might not in principle provide useful information. Moderating this statement might perhaps be considered. Other than that, and in the assumed absence of hypothetical flaws that a specialist on metallic materials might identify, I see no problem with publishing this response.

That said, the point of broader interest here is not the details of metallic engineering, but what has rather misleadingly become known as “the hypothesis of Lucas et al.”. I would like to use this opportunity to revisit the original findings reported by them and relate them to the current disagreement. It seems to have been perceived by most readers that Lucas et al. actually established, or at least claimed, that phytoliths do not wear dental enamel. This is a misapprehension. In my view the key sentence in their abstract is not “We conclude that dust has overwhelming importance as a wear agent and that dietary signals preserved in dental microwear are indirect” but “By contrast, phytoliths and enamel chips deformed during sliding, forming U-shaped grooves or flat troughs in enamel, without tissue loss.” This is further elucidated in the discussion: “Contacts with these [phytolith] and other particles would, at most, merely rearrange the enamel surface via rubbing. While such rubbing may eventually lead to wear, many further contacts are required before cracks detach the tissue”. It is not entirely clear what “many” means in this setting, but for a horse chewing hay at a rate of about once per second this could potentially amount to several thousand contacts per hour, tens of thousands per day and hundreds of millions per lifetime. In other words, Lucas et al. (2013) unequivocally and for the first time showed that squash phytoliths did have an effect on orangutang enamel and even, if indirectly, suggested that actual wear would result under realistic scenarios of chewing. How in a living animal this relates to the more severe wear induced per item by quartz particles depends on many factors, including the relative amounts of different kinds of particles actually present by the time the food is being masticated.

It is regrettable that the careful discussion of the processes of dental wear in relation to the properties of wear-inducing particles opened by Lucas et al. (2013) has received so little attention. One would hope that the discussion could soon move away from infertile quibble about relative hardness to the richer perspective of considering the interaction of different kinds of particles in the complicated system of food being transported across and deformed between dental surfaces of various shapes and properties. The concluding words of Lucas et al. put this well: “Much needs to be done. On the theoretical side, variation of the critical attack angle β with indenter geometry requires investigation. Further research is needed to extend the theory to include conditions of mutual wear. On the practical side, the work needs to be extended to dentine. Phytoliths would form rigid-plastic contacts with this tissue because dentinal hardness is generally 650 MPa or less. This would influence the enamel–dentine wear ratio in teeth, where both tissues are exposed, in a manner not previously contemplated.” While much could be

learned from feeding experiments with live animals there are serious limits to what can actually be done, practical as well as ethical. I do not doubt that experimental work using properly designed mechanical models, possibly including metallic materials, could be very helpful here.

Reference

Lucas PW et al. 2013 Mechanisms and causes of wear in tooth enamel: implications for hominin diets. *J. R. Soc. Interface* 10, 20120923. (doi:10.1098/rsif.2012.0923)

Review form: Reviewer 2

Is the manuscript scientifically sound in its present form?

Yes

Are the interpretations and conclusions justified by the results?

Yes

Is the language acceptable?

Yes

Is it clear how to access all supporting data?

Not Applicable

Do you have any ethical concerns with this paper?

No

Have you any concerns about statistical analyses in this paper?

No

Recommendation?

Accept as is

Comments to the Author(s)

An extremely clear and well written response. I am convinced and I have little to say. In my view, the authors should avoid substituting proper nouns with reference numbers:

Page 3, line 39 is better as "Xia et al. [4] cite Hernot et al. [6]"

Page 3, line 46 is better as "addressed by Hernot et al. [6]"

Page 3, line 48 is better as "used by van Casteren et al. [3]"

Page 4, line 19 is better as "that used by van Casteren et al. [3]"

Decision letter (RSOS-190572.R0)

21-May-2019

Dear Dr van Casteren

On behalf of the Editors, I am pleased to inform you that your Manuscript RSOS-190572 entitled "Metallic proxies remain unsuitable for assessing the mechanics of microwear formation: reply to

comment on van Casteren et al. (2018)" has been accepted for publication in Royal Society Open Science subject to minor revision in accordance with the referee suggestions. Please find the referees' comments at the end of this email.

The reviewers and handling editors have recommended publication, but also suggest some minor revisions to your manuscript. Therefore, I invite you to respond to the comments and revise your manuscript.

- Ethics statement

- Data accessibility

If you wish to submit your supporting data or code to Dryad (<http://datadryad.org/>), or modify your current submission to dryad, please use the following link:
<http://datadryad.org/submit?journalID=RSOS&manu=RSOS-190572>

- Competing interests

- Authors' contributions

- Acknowledgements

- Funding statement

Because the schedule for publication is very tight, it is a condition of publication that you submit the revised version of your manuscript before 30-May-2019. Please note that the revision deadline will expire at 00.00am on this date. If you do not think you will be able to meet this date please let me know immediately.

Supplementary files will be published alongside the paper on the journal website and posted on the online figshare repository (<https://rs.figshare.com/>). The heading and legend provided for each supplementary file during the submission process will be used to create the figshare page,

so please ensure these are accurate and informative so that your files can be found in searches. Files on figshare will be made available approximately one week before the accompanying article so that the supplementary material can be attributed a unique DOI.

on behalf of Dr Jake Socha (Associate Editor) and Kevin Padian (Subject Editor)
openscience@royalsociety.org

Associate Editor Comments to Author (Dr Jake Socha):

Associate Editor: 1

Comments to the Author:

The reviewers are both positive, and their reviews have provided additional comments and insight worthy of consideration. Reviewer 1 gives an outside perspective on the debate; I strongly encourage you to think it over and revise/amend the letter where appropriate, and if possible.

Reviewer comments to Author:

Reviewer: 1

Comments to the Author(s)

This is a hard one to review, mostly because the discussion has moved into a highly specialised realm of metallic engineering where few will be able to assess the arguments with confidence. It might be tempting to agree with the punch line that “sliding experiments using metallic balls remain irrelevant to testing the hypothesis of Lucas et al.” – but while there are clearly technical complications to be discussed it is by no means obvious that such experiments might not in principle provide useful information. Moderating this statement might perhaps be considered. Other than that, and in the assumed absence of hypothetical flaws that a specialist on metallic materials might identify, I see no problem with publishing this response.

That said, the point of broader interest here is not the details of metallic engineering, but what has rather misleadingly become known as “the hypothesis of Lucas et al.”. I would like to use this

opportunity to revisit the original findings reported by them and relate them to the current disagreement. It seems to have been perceived by most readers that Lucas et al. actually established, or at least claimed, that phytoliths do not wear dental enamel. This is a misapprehension. In my view the key sentence in their abstract is not "We conclude that dust has overwhelming importance as a wear agent and that dietary signals preserved in dental microwear are indirect" but "By contrast, phytoliths and enamel chips deformed during sliding, forming U-shaped grooves or flat troughs in enamel, without tissue loss." This is further elucidated in the discussion: "Contacts with these [phytolith] and other particles would, at most, merely rearrange the enamel surface via rubbing. While such rubbing may eventually lead to wear, many further contacts are required before cracks detach the tissue". It is not entirely clear what "many" means in this setting, but for a horse chewing hay at a rate of about once per second this could potentially amount to several thousand contacts per hour, tens of thousands per day and hundreds of millions per lifetime. In other words, Lucas et al. (2013) unequivocally and for the first time showed that squash phytoliths did have an effect on orangutang enamel and even, if indirectly, suggested that actual wear would result under realistic scenarios of chewing. How in a living animal this relates to the more severe wear induced per item by quartz particles depends on many factors, including the relative amounts of different kinds of particles actually present by the time the food is being masticated.

It is regrettable that the careful discussion of the processes of dental wear in relation to the properties of wear-inducing particles opened by Lucas et al. (2013) has received so little attention. One would hope that the discussion could soon move away from infertile quibble about relative hardness to the richer perspective of considering the interaction of different kinds of particles in the complicated system of food being transported across and deformed between dental surfaces of various shapes and properties. The concluding words of Lucas et al. put this well: "Much needs to be done. On the theoretical side, variation of the critical attack angle θ with indenter geometry requires investigation. Further research is needed to extend the theory to include conditions of mutual wear. On the practical side, the work needs to be extended to dentine. Phytoliths would form rigid-plastic contacts with this tissue because dentinal hardness is generally 650 MPa or less. This would influence the enamel-dentine wear ratio in teeth, where both tissues are exposed, in a manner not previously contemplated." While much could be learned from feeding experiments with live animals there are serious limits to what can actually be done, practical as well as ethical. I do not doubt that experimental work using properly designed mechanical models, possibly including metallic materials, could be very helpful here.

Reference

Lucas PW et al. 2013 Mechanisms and causes of wear in tooth enamel: implications for hominin diets. *J. R. Soc. Interface* 10, 20120923. (doi:10.1098/rsif.2012.0923)

Reviewer: 2

Comments to the Author(s)

An extremely clear and well written response. I am convinced and I have little to say. In my view, the authors should avoid substituting proper nouns with reference numbers:

Page 3, line 39 is better as "Xia et al. [4] cite Hernot et al. [6]"

Page 3, line 46 is better as "addressed by Hernot et al. [6]"

Page 3, line 48 is better as "used by van Casteren et al. [3]"

Page 4, line 19 is better as "that used by van Casteren et al. [3]"

Author's Response to Decision Letter for (RSOS-190572.R0)

See Appendix A.

Decision letter (RSOS-190572.R1)

13-Jun-2019

Dear Dr van Casteren:

On behalf of the Editors, I am pleased to inform you that your Manuscript RSOS-190572.R1 entitled "Metallic proxies remain unsuitable for assessing the mechanics of microwear formation: reply to comment on van Casteren et al. (2018)" has been accepted for publication in Royal Society Open Science subject to minor revision in accordance with the editor's suggestions. Please find the editor's comments at the end of this email.

The Subject Editor has recommended publication, but also suggests some minor revisions to your manuscript. Therefore, I invite you to respond to the comments and revise your manuscript.

- Ethics statement

- Data accessibility

If you wish to submit your supporting data or code to Dryad (<http://datadryad.org/>), or modify your current submission to dryad, please use the following link:
<http://datadryad.org/submit?journalID=RSOS&manu=RSOS-190572.R1>

- Competing interests

- Authors' contributions

All submissions, other than those with a single author, must include an Authors' Contributions section which individually lists the specific contribution of each author. The list of Authors should meet all of the following criteria; 1) substantial contributions to conception and design, or

acquisition of data, or analysis and interpretation of data; 2) drafting the article or revising it critically for important intellectual content; and 3) final approval of the version to be published.

- Acknowledgements

- Funding statement

Because the schedule for publication is very tight, it is a condition of publication that you submit the revised version of your manuscript before 22-Jun-2019. Please note that the revision deadline will expire at 00.00am on this date. If you do not think you will be able to meet this date please let me know immediately.

- 1) A text file of the manuscript (tex, txt, rtf, docx or doc), references, tables (including captions) and figure captions. Do not upload a PDF as your "Main Document".
- 2) A separate electronic file of each figure (EPS or print-quality PDF preferred (either format should be produced directly from original creation package), or original software format)
- 3) Included a 100 word media summary of your paper when requested at submission. Please ensure you have entered correct contact details (email, institution and telephone) in your user account

- 4) Included the raw data to support the claims made in your paper. You can either include your data as electronic supplementary material or upload to a repository and include the relevant doi within your manuscript
- 5) All supplementary materials accompanying an accepted article will be treated as in their final form. Note that the Royal Society will neither edit nor typeset supplementary material and it will be hosted as provided. Please ensure that the supplementary material includes the paper details where possible (authors, article title, journal name).

on behalf of Dr Jake Socha (Associate Editor) and Kevin Padian (Subject Editor)
openscience@royalsociety.org

Associate Editor Comments to Author (Dr Jake Socha):

Please add a space between numbers and units. Aside from this minor comment, everything else looks good; thank you for addressing the comments of the reviewers. After publication of this letter, we will close the books on this thread, and look forward to seeing future research papers on the topic.

Lastly, could you change figure 2b so that the y axis label reads "Average pressure (GPa)" rather than "Av.Pressure (GPa)?"

Author's Response to Decision Letter for (RSOS-190572.R1)

See Appendix B.

Decision letter (RSOS-190572.R2)

19-Jun-2019

Dear Dr van Casteren,

I am pleased to inform you that your manuscript entitled "Metallic proxies remain unsuitable for assessing the mechanics of microwear formation: reply to comment on van Casteren et al. (2018)" is now accepted for publication in Royal Society Open Science.

Kind regards,
Lianne Parkhouse
Royal Society Open Science
openscience@royalsociety.org

on behalf of Dr Jake Socha (Associate Editor) and Kevin Padian (Subject Editor)
openscience@royalsociety.org

Appendix A

Dear Editors,

We thank you very much for the consideration of our paper and we are grateful to the reviewers for their helpful comments. We have revised the manuscript in accord with both reviewer and editorial comments, interpolating below in blue the specific changes that we have been made.

We especially grateful to Reviewer 1 for the thorough and thoughtful perspective on the topic and like the reviewer we look forward to future experimental work using properly designed mechanical models to elucidate the tribology of dental enamel.

All the best

Adam van Casteren

Associate Editor Comments to Author (Dr Jake Socha):

Associate Editor: 1

Comments to the Author:

The reviewers are both positive, and their reviews have provided additional comments and insight worthy of consideration. Reviewer 1 gives an outside perspective on the debate; I strongly encourage you to think it over and revise/amend the letter where appropriate, and if possible.

Reviewer comments to Author:

Reviewer: 1

Comments to the Author(s)

This is a hard one to review, mostly because the discussion has moved into a highly specialised realm of metallic engineering where few will be able to assess the arguments with confidence. It might be tempting to agree with the punch line that “sliding experiments using metallic balls remain irrelevant to testing the hypothesis of Lucas *et al.*” – but while there are clearly technical complications to be discussed it is by no means obvious that such experiments might not in principle provide useful information. Moderating this statement might perhaps be considered. Other than that, and in the assumed absence of hypothetical flaws that a specialist on metallic materials might identify, I see no problem with publishing this response.

The final sentence has been moderated and now reads

“In sum, sliding experiments using metallic balls present little useful data for testing the hypothesis of Lucas *et al.* [1].”

That said, the point of broader interest here is not the details of metallic engineering, but what has rather misleadingly become known as “the hypothesis of Lucas *et al.*”. I would like to use this opportunity to revisit the original findings reported by them and relate them to the current disagreement. It seems to have been perceived by most readers that Lucas *et al.* actually established, or at least claimed, that phytoliths do not wear dental enamel. This is a misapprehension. In my view the key sentence in their abstract is not “We conclude that dust has overwhelming importance as a wear agent and that dietary signals preserved in dental microwear are indirect” but “By contrast, phytoliths and enamel chips deformed during sliding, forming U-shaped grooves or flat troughs in enamel, without tissue loss.” This is further elucidated in the

discussion: "Contacts with these [phytolith] and other particles would, at most, merely rearrange the enamel surface via rubbing. While such rubbing may eventually lead to wear, many further contacts are required before cracks detach the tissue". It is not entirely clear what "many" means in this setting, but for a horse chewing hay at a rate of about once per second this could potentially amount to several thousand contacts per hour, tens of thousands per day and hundreds of millions per lifetime. In other words, Lucas et al. (2013) unequivocally and for the first time showed that squash phytoliths did have an effect on orangutang enamel and even, if indirectly, suggested that actual wear would result under realistic scenarios of chewing. How in a living animal this relates to the more severe wear induced per item by quartz particles depends on many factors, including the relative amounts of different kinds of particles actually present by the time the food is being masticated.

It is regrettable that the careful discussion of the processes of dental wear in relation to the properties of wear-inducing particles opened by Lucas et al. (2013) has received so little attention. One would hope that the discussion could soon move away from infertile quibble about relative hardness to the richer perspective of considering the interaction of different kinds of particles in the complicated system of food being transported across and deformed between dental surfaces of various shapes and properties. The concluding words of Lucas et al. put this well: "Much needs to be done. On the theoretical side, variation of the critical attack angle α with indenter geometry requires investigation. Further research is needed to extend the theory to include conditions of mutual wear. On the practical side, the work needs to be extended to dentine. Phytoliths would form rigid-plastic contacts with this tissue because dentinal hardness is generally 650 MPa or less. This would influence the enamel–dentine wear ratio in teeth, where both tissues are exposed, in a manner not previously contemplated." While much could be learned from feeding experiments with live animals there are serious limits to what can actually be done, practical as well as ethical. I do not doubt that experimental work using properly designed mechanical models, possibly including metallic materials, could be very helpful here.

Reference

Lucas PW et al. 2013 Mechanisms and causes of wear in tooth enamel: implications for hominin diets. *J. R. Soc. Interface* 10, 20120923. (doi:10.1098/rsif.2012.0923)

We thank the reviewer for this thoughtful and detailed opinion, and we agree that further well-designed experimentation will progress the understanding of dental tribology. However, where possible, we believe the incorporation of biologically derived materials rather than proxy materials would better serve this purpose.

Reviewer: 2

Comments to the Author(s)

An extremely clear and well written response. I am convinced and I have little to say. In my view, the authors should avoid substituting proper nouns with reference numbers:

Many thanks to the reviewer

Page 3, line 39 is better as "Xia et al. [4] cite Hernot et al. [6]"

Page 3, line 46 is better as "addressed by Hernot et al. [6]"

Page 3, line 48 is better as "used by van Casteren et al. [3]"

Page 4, line 19 is better as "that used by van Casteren et al. [3]"

The above changes have been made

Appendix B

Dear Dr. Jake Socha,

We are delighted the reply has been accepted and have revised the manuscript in accord with your editorial comments, interpolating below in blue the specific changes that we have been made.

All the best

Adam van Casteren

Please add a space between numbers and units. Aside from this minor comment, everything else looks good; thank you for addressing the comments of the reviewers. After publication of this letter, we will close the books on this thread, and look forward to seeing future research papers on the topic.

Throughout the document a space has been added between numbers and units.

Lastly, could you change figure 2b so that the y axis label reads “Average pressure (GPa)” rather than “Av.Pressure (GPa)?

Y-axis label in figure 2b has been changed. We have also corrected the Y-axis in figure 1a that was erroneously labelled “(mm²)” it has now been corrected to “(nm²)”.